

# Asymmetries in mother-infant behaviour in Barbary macaques (*Macaca sylvanus*)

Barbara Regaiolli[1], Caterina Spiezio[1] and William Donald Hopkins[2]

[1] Research & Conservation Department, Parco Natura Viva—Garda Zoological Park, Verona, Italy
[2] Neuroscience Institute and Language Research Center, Georgia State University, Atlanta, GA, United States of America

Corresponding author
Barbara Regaiolli,
barbara.regaiolli@parconaturaviva.it

## ABSTRACT

Asymmetries in the maternal behaviour and anatomy might play an important role in the development of primate manual lateralization. In particular, early life asymmetries in mother's and infant's behaviour have been suggested to be associated with the development of the hand preference of the offspring. The aim of this study was to investigate the presence of behavioural asymmetries in different behavioural categories of mother-infant dyads of zoo-living Barbary macaques (*Macaca sylvanus*). The study subjects were 14 Barbary macaques involved in seven mother-infant dyads housed in Parco Natura Viva, Italy. For the mothers, bouts of hand preference for maternal cradling and infant retrieval were collected. For the infants, we focused on nipple preference and hand preference for clinging on mother ventrum. Moreover, we collected bouts of hand preference for food reaching in both groups. No significant group-level bias was found for any of the behavioural categories in either mothers or infants. However, at the individual level, six out of seven mothers showed a significant cradling bias, three toward the right hand and three toward the left hand. Moreover, all infants showed a significant nipple preference, six toward the mother's right nipple, one toward the left nipple. Furthermore, a significant correlation was found between the infant nipple preference and their hand preference for food reaching, suggesting that maternal environment rather than behaviour might affect the development of hand preference in Old World monkeys. Our findings seem partially to add to previous literature on perceptual lateralization in different species of non-primate mammals, reporting a lateral bias in mother-infant interactions. Given the incongruences between our study and previous research in great apes and humans, our results seem to suggest possible phylogenetic differences in the lateralization of mothers and infants within the *Primates* order.

## INTRODUCTION

The lateralization in mother-infant interactions has recently been hypothesised to have a perceptual origin in different mammal species, from non-primate mammals (*Karenina et al., 2017*) to humans (*Todd & Banerjee, 2016*). In particular, holding infant on the left side of the body has been found to enhance the mother visual and tactile monitoring skill, due to the involvement of the right hemisphere of the brain which is specialised for attention (*Whitehead, 1991*; *Bourne & Todd, 2004*; *Todd & Banerjee, 2016*) and tactile

information (*Coghill, Gilron & Iadarola, 2001*; *Sakai et al., 2006*). Among mammals the right hemisphere has been found to be more involved in social processing, such as monitoring conspecifics and attending social responses (*Rogers, Vallortigara & Andrew, 2013*; *Quaresmini et al., 2014*; *Giljov, Karenina & Malashichev, 2018*). Moreover, recent studies reported pronounced lateralization of mother-infant behaviour in several non-primate mammals, indicating right-hemisphere dominance for social processing in species of bovids, equids, macropods and marine mammals (*Karenina et al., 2017*; *Giljov, Karenina & Malashichev, 2018*). In particular, in all these species infants have been found to prefer keeping the mothers on the left side, compared with the right side, adding further evidence that lateralized mother-infant interactions emerged long before primates (*Karenina et al., 2017*; *Giljov, Karenina & Malashichev, 2018*). This left side-right hemisphere preference in the infants has been hypothesised to be advantageous for their survival, favouring mother-infant bonding behaviour and maintenance of spatial proximity (*Karenina et al., 2017*). Regarding humans, the left-cradling bias found in our species has been hypothesized to be related for example to the infant imprinting to the mother heartbeat, with left-cradling allowing the infant to be close to the mother's heart (*Salk, 1973*). Thus, in the presence of an emotional involvement both humans and non-primate mammals seem to choose keeping the conspecifics on their left side. Indeed, the right hemisphere is competent in processing the visuo-spatial information, resulting in an attentional bias toward the left visual hemifield (*Karenina et al., 2017*). This view has been supported also in other studies of social affiliative behaviour. For example, a left bias for embraces related to positive emotions, a kind of adult cradling, has been reported in humans (*Packheiser et al., 2018*) and spider monkeys (*Boeving, Belnap & Nelson, 2017*). According to recent research, the right hemisphere involvement in social stimuli control has also been hypothesized to be a reason for the left-cradling bias reported in the maternal behaviour of humans and some great apes (*Hopkins, 2004*; *Rosa Salva et al., 2012*; *Giljov, Karenina & Malashichev, 2018*). Indeed, research on humans revealed that mothers prefer to cradle their infant on the left side (*Salk, 1960*; *Damerose & Vauclair, 2002*) and similar findings have been reported in chimpanzees (*Pan troglodytes*) (*Manning & Chamberlain, 1990*; *Hopkins et al., 1993*; *Manning, Heaton & Chamberlain, 1994*; E Toback, 1999, unpublished data) and gorillas (*Gorilla gorilla*) (*Manning, Heaton & Chamberlain, 1994*) (Table 1). On the other hand, asymmetries in maternal cradling seem to be less clear in Old World monkeys. Indeed, no group-level bias was found for this behaviour in Japanese macaques (*Macaca fuscata*) (*Tanaka, 1989*), rhesus macaques (*Macaca mulatta*) (*Tomaszycki et al., 1998*), olive baboons (*Papio anubis*) (*Damerose & Hopkins, 2002*) and Sichuan snub-nose monkeys (*Rhinopithecus roxellana*) (*Zhao et al., 2008*) (Table 1), suggesting phylogenetic differences between taxonomic primate groups. In Old World monkeys, the lack of lateral biases has been found also in other kind of mother-infant interactions, particularly infant retrieval. Indeed, research on rhesus macaques and olive baboons did not report any group-level side preferences in the hand used to retrieve the infant by the mother (*Tomaszycki et al., 1998*; *Damerose & Hopkins, 2002*) (Table 1).

Infant nipple preference has been investigated in different non-human primate species, revealing a bias toward the left nipple in wild chimpanzees (*Nishida, 1993*), captive

**Table 1  Previous studies on lateralization in mother-infant interactions.** The table provides a summary of previous research carried out with infants or mother-infant dyads of Old World monkeys and great apes. Each row reports a different study, showing the authors and year of publication, the species (Latin name) and group-level statistical results about lateralization in maternal cradling, retrieval of the infant by the mothers (infant retrieval) and preference of the infant for the mother's nipple (nipple preference).

| Author and publication year | Species | Context | Reported bias |
| --- | --- | --- | --- |
| **Old-World monkeys** | | | |
| *Hiraiwa, 1981* | *Macaca fuscata* | Nipple preference | No bias |
| *Tanaka, 1989* | *Macaca fuscata* | Maternal cradling | No bias |
| | | Nipple preference | No bias |
| *Tomaszycki et al., 1998* | *Macaca mulatta* | Maternal cradling | No bias |
| | | Infant retrieval | No bias |
| | | Nipple preference | Left |
| *Jaffe et al., 2006* | *Macaca mulatta (wild)* | Nipple preference | No bias. |
| *Lindburg, 1971* | *Macaca mulatta (wild)* | Nipple preference | Right |
| *Erwin, Anderson & Bunger, 1975* | *Macaca nemestrina* | Nipple preference | No bias. |
| *Damerose & Vauclair, 2002* | *Papio anubis* | Maternal cradling | No bias |
| | | Infant retrieval | No bias |
| | | Nipple preference | No bias |
| *Zhao et al., 2008* | *Rhinopithecus roxellana* | Maternal cradling | No bias |
| | | Nipple preference | No bias |
| **Great apes** | | | |
| *Manning, Heaton & Chamberlain, 1994* | *Gorilla gorilla* | Maternal cradling | Left |
| *Hopkins & De Lathouwers, 2006* | *Pan paniscus* | Nipple preference | Left |
| *Hopkins & De Lathouwers, 2006* | *Pan troglodytes* | Nipple preference | Left |
| *Hopkins et al., 1993* | *Pan troglodytes* | Maternal cradling | Left |
| *Manning & Chamberlain, 1990* | *Pan troglodytes* | Maternal cradling | Left |
| *Manning, Heaton & Chamberlain, 1994* | *Pan troglodytes* | Maternal cradling | Left |
| E Toback, 1999, unpublished data | *Pan troglodytes* | Maternal cradling | Left |
| *Nishida, 1993* | *Pan troglodytes (wild)* | Nipple preference | Left |

chimpanzees and bonobos (*Pan paniscus*) (*Hopkins & De Lathouwers, 2006*). As reported for maternal cradling, in general no group-level bias in nipple preference has been reported in past research on Old World monkeys, particularly Japanese macaques (*Hiraiwa, 1981*; *Tanaka, 1989*), pig-tailed macaques (*Macaca nemestrina*) (*Erwin, Anderson & Bunger, 1975*), olive baboons (*Damerose & Hopkins, 2002*) and Sichuan snub-nose monkeys (*Zhao et al., 2008*). On the other hand, wild rhesus macaques have been found to show a right nipple preference (*Lindburg, 1971*) whereas an opposite bias has been reported in a captive group of this species, showing a slight group-level left nipple preference (*Tomaszycki et al., 1998*). In contrast, more recent research on a large troop of wild rhesus macaques found no group-level nipple preference in this species (*Jaffe et al., 2006*) (Table 1).

Research on hand preference on different tasks in monkeys and, to a lesser extent, great apes has given rise to a heterogeneous picture of their manual lateralization (for review: *Papademetriou, Sheu & Michel, 2005*; *Fitch & Braccini, 2013*). The inconsistency

between different studies might be due to methodological differences as well as to the potential influence of factors such as posture (*Forsythe et al., 1988*; *MacNeilage, 2007*), task (*Fagot & Vauclair, 1991*), and individual experience and learning (*Westergaard & Suomi, 1993*; *Hopkins, 1995*; *Meunier, Blois-Heulin & Vauclair, 2011*). One of the main hypotheses aiming to explain patterns of hand preference in primates is the Postural Origin Theory (POT) by *MacNeilage (2007)*. According to the POT, the adoption of terrestrial habits during primate evolution allowed the right hand to become free from postural support duties, getting gradually more involved in tasks requiring specific skill. This process resulted in the pronounced right handedness characterizing humans with their bipedal posture (*MacNeilage, 2007*; *Meguerditchian, Vauclair & Hopkins, 2013*; *Blois-Heulin et al., 2006*). In addition, also the asymmetries in the intrauterine environment as well as in the maternal behaviour and anatomy might play an important role in the development of primate manual lateralization (*Previc, 1991*; *Hepper, Shahidullah & White, 1991*; *Hopkins, 2004*). In fact, basing on previous research, early life asymmetries in mothers and infants have been suggested to be associated with the development of the hand preference of the offspring (*Hopkins, 1994*; *Hopkins, 1995*; *Westergaard, Byrne & Suomi, 1998*; *Hopkins, 2004*). Moreover, mothers and infant might reciprocally impact their manual lateralization (*Scola & Vauclair, 2010*). Investigating behavioural asymmetries in mothers and infants might therefore be useful to better understand the development of motor lateralization, particularly handedness in non-human primates, gaining new insight into factors driving the evolution of manual laterality in these species (*Hopkins, 2004*). However, although literature on humans and chimpanzees is relatively well-represented, more studies are needed involving monkey and prosimian mother-infant dyads (*Hopkins, 2004*).

The aim of the current study was to verify the presence of behavioural asymmetries in mother-infant dyads in a sample of zoo Barbary macaques (*Macaca sylvanus)*, investigating whether mother lateralization correlates with that of the infant. In particular, for mother macaques we assessed lateralization in maternal cradling and infant retrieval, whereas for the infants we focused on nipple preference and hand use to cling on the mother ventrum. In addition, we tested and investigated the relationship between mother and infant hand preference for food reaching. Basing on previous literature on Old World monkeys, specifically macaques, we expect no bias in maternal cradling and infant retrieval by the mothers (*Tomaszycki et al., 1998*; *Damerose & Hopkins, 2002*).

## METHODS

### Study subjects and area

The study was carried out with 14 Barbary macaques involved in seven mother-infant dyads. The study dyads lived in a multimale-multifemale social group composed of 24 macaques: a dominant male, four sub-adult males, five one-year old juveniles and the seven adult females with their offspring (seven infants in total) involved in the study. All subjects were housed at Parco Natura Viva, a zoological garden in Verona (Italy). The macaques were housed in an 870 m$^2$ naturalistic enclosure made of grassy areas, plants and trees, rocks, high ropes, artificial shelters and a water pool. Barbary macaques were fed twice

a day and water was available to the animals ad libitum. The diet consisted of fruits and vegetables, seeds, cereals and mealworms. On a daily basis, macaques were involved in an environmental enrichment program and could receive manipulative devices to be opened to reach for food as well as foraging enrichment. In this latter case, small pieces of food were scattered around in the enclosure or hidden in hay or straw mounds.

All subjects of the study were born in captivity and were parent-reared. The study was carried out through the live observation of spontaneous behaviours of macaques in their social context. No invasive or stressful techniques were used, and the study procedure was in accordance with the EU Directive 2010/63/EU for animal research and the Italian legislative decree 26/2014 for Animal Research.

## Procedure and data collection

The study was carried when the infants were between 20 to 30 days of age. For each mother and for each infant, fourteen 15-minute sessions were carried out. In particular, two sessions per day were done, one in the morning, one in the afternoon. A continuous focal animal sampling method was used to collect the bouts of right and left-hand use for different behavioural categories in both mothers and infants and to collect the bouts of infant nipple preference. A bout was intended as the first event of a series and we did not record more than one response if it was not separated by a behavioural event or postural change of the macaque (*McGrew & Marchant, 1997*; *Hopkins et al., 2001*; *Regaiolli, Spiezio & Hopkins, 2018*). Regarding the mothers, data on the hand preference for maternal cradling and infant retrieval were collected. Maternal cradling was defined as the mother holding the offspring ventrally (*Damerose & Hopkins, 2002*). Retrieval was defined as the mother reaching to retrieve an infant not in contact with her for any reason. In particular, we collected only unambiguous bouts of cradling and infant retrieval performed with only the right or the left hand, whereas bouts carried out with both hands were not considered. Regarding the infants, data on the nipple preference and the hand preference for clinging on the mother's ventrum were collected. For nipple preference, we recorded data of suckling on the mother right and left nipple. In particular, we collected suckling bouts, intended as the first event of a series: one bout started when the infant's mouth was put on the mother's nipple and ended when the mouth was removed from it, with no distinction between different suckling phases (*Damerose & Hopkins, 2002*). For clinging on the mothers ventrum, we recorded the hand used by the infant to hold to the mother fur, while the other hand was doing a different action or was not involved in any activity. In addition, for both mothers and infants, data on the hand used to reach for food (referred to as "food reaching" throughout the manuscript) in the feeding points of the enclosure were collected. In particular, only unambiguous bouts performed with one hand were considered. Bimanual reaching, ambiguous bouts and reaching from asymmetrical postures (e.g.: laying on the side) were not included. For the infants, we considered only those reaching bouts that took place when the subject was on the ground and no hand was in contact with the mother.

## Data analysis

Individual-level lateralization was evaluated using binomial $z$-scores, a similar methodology to that carried out in previous studies of hand preference in non-human primates (*Michel, Sheu & Brumley, 2002*; *Hopkins et al., 2004*; *Llorente et al., 2011*; *Hashimoto, Yamazaki & Iriki, 2013*). In particular, the critical values of the $z$-scores were set at $-1.96$ and $1.96$, associated with a $p$-value of 0.05. The $z$-scores were used to classify the subjects as left-handed ($z < -1.96$), right-handed ($z > 1.96$) and ambi-preferent ($-1.96 < z < 1.96$). For each behavioural category considered in the study, only subjects that reached a minimum of ten bouts were included in the individual-level analysis (*Meguerditchian, Vauclair & Hopkins, 2010*). Moreover, for each subject a Laterality Index score (LI) was calculated. The LI was given by the formula LI = $(R - L)/(R + L)$; R stands for the frequencies of the right side/hand use and L stands for the frequencies of the left side/hand use. The LI varies between $-1.0$ and $1.0$ with negative values indicating a left side bias, whereas positive values indicate a right-side bias (e.g., *Hopkins & De Lathouwers, 2006*; *Zhao et al., 2008*; *Giljov, Karenina & Malashichev, 2018*). Moreover, to compare the strength of the hand preference for food reaching between mother and infants, the absolute values of the LI (ABS-LI) were considered. Given that Kolmogorov–Smirnov Goodness-of-Fit tests revealed that not all data were normally distributed, non-parametric statistical tests were used for the group-level analyses. In particular, group-level side or hand preferences were evaluated using a one-sample sign-test with the Laterality Index serving as dependent measure and chance level was set at 0. For the mothers, the LI and ABS-LI of cradling, infant retrieval and food reaching (behavioural categories that involved the use of one hand) were compared using the Wilcoxon test. Similarly, for the infants, Wilcoxon test was used to compare clinging on mother ventrum and food reaching. Moreover, a Spearman correlation was run on the LI to investigate the relationship between infants' nipple preference and their hand preference for food reaching. Finally, to compare handedness between mother and infants, we compared the LI and the ABS-LI score for food reaching between the two groups using a Mann–Whitney test. All tests were two-tailed. The significance level was set at $p < 0.05$. When performing Wilcoxon tests for multiple comparisons, the $p$-value of 0.05 was adjusted with the Bonferroni-Holm correction (*Holm, 1979*).

# RESULTS

## Maternal cradling and infant retrieval

The median LI (IQR-Interquartile range, minimum, maximum) for maternal cradling was $-0.18$ (0.91, $-0.69$, 0.72). At the individual level, six out of seven subjects showed a significant cradling bias, three toward the left hand and three toward the right hand (Table 2). According to a one-sample sign-test, no bias in the distribution of the LI was found ($p = 1$, $N = 7$) (Fig. 1). The median LI (IQR, minimum, maximum) for infant retrieval was 0 (0.18, $-0.14$, 0.22). At the individual level, no macaque showed a significant lateralization (Table 2) and, according to a one-sample sign-test, no bias in the distribution of the LI was found ($p = 0.453$, $N = 7$) (Fig. 1).

**Table 2  Measures of lateral biases in mother and infant Barbary macaques.**

| Subject | Maternal cradling | | | | | Infant retrieval | | | | |
|---|---|---|---|---|---|---|---|---|---|---|
| | Right (bouts) | Left (bouts) | Laterality index | z-score | Preference | Right (bouts) | Left (bouts) | Laterality Index | z-score | Preference |
| Budda | 101 | 34 | 0.50 | 5.68 | Right | 13 | 11 | 0.08 | 0.2 | Ambi |
| Funny | 51 | 77 | −0.20 | −2.21 | Left | 11 | 11 | 0.00 | 0 | Ambi |
| Katrina | 5 | 27 | −0.69 | −3.71 | Left | 3 | 4 | −0.14 | [a] | [a] |
| Last | 11 | 51 | −0.65 | −4.95 | Left | 11 | 7 | 0.22 | 0.71 | Ambi |
| Mirror | 49 | 71 | −0.18 | −1.92 | Ambi | 18 | 12 | 0.20 | 0.91 | Ambi |
| Vanda | 98 | 16 | 0.72 | 7.59 | Right | 14 | 14 | 0.00 | 0 | Ambi |
| Violetta | 74 | 27 | 0.47 | 4.58 | Right | 24 | 28 | −0.08 | −0.42 | Ambi |

| | Infant nipple preference | | | | | Clinging on ventrum | | | | |
|---|---|---|---|---|---|---|---|---|---|---|
| | Right (bouts) | Left (bouts) | Laterality index | z-score | Preference | Right (bouts) | Left (bouts) | Laterality Index | z-score | Preference |
| Budda's Infant | 66 | 1 | 0.97 | 7.82 | Right | 121 | 143 | −0.08 | −1.29 | Ambi |
| Funny's Infant | 63 | 8 | 0.77 | 6.41 | Right | 122 | 128 | −0.02 | −0.32 | Ambi |
| Katrina's Infant | 26 | 6 | 0.63 | 3.36 | Right | 96 | 86 | 0.05 | 0.67 | Ambi |
| Last's Infant | 69 | 6 | 0.84 | 7.16 | Right | 122 | 102 | 0.09 | 1.27 | Ambi |
| Mirror's Infant | 32 | 7 | 0.64 | 3.84 | Right | 143 | 149 | −0.02 | −0.29 | Ambi |
| Vanda's Infant | 51 | 26 | 0.32 | 2.74 | Right | 107 | 115 | −0.04 | −0.47 | Ambi |
| Violetta's Infant | 11 | 46 | −0.61 | −4.50 | Left | 159 | 169 | −0.03 | −0.5 | Ambi |

**Notes.**
[a]Excluded from the individual-level statistical analysis due to data deficiency.
Ambi: −1.96 < z-score < 1.96; Left: z-score < −1.96; Right: z-score > 1.96.

### Nipple preference and clinging on the mother's ventrum

Regarding infant nipple preference, the median LI (IQR, minimum, maximum) was 0.64 (0.33, −0.61, 0.97). At the individual level, all subjects showed a significant nipple preference, one toward the mother's left nipple and six toward the right nipple (Table 2). According to a one-sample sign-test, no bias in the distribution of the LI was found ($p = 0.125$, $N = 7$) (Fig. 1). In the case of the hand preference for clinging on the mother's ventrum, the median LI (IQR, minimum, maximum) was −0.02 (0.06, −0.08, 0.09). At the individual level, no macaque showed a significant lateralization (Table 2). One-sample sign-test revealed no bias in the distribution of the LI ($p = 0.453$, $N = 7$) (Fig. 1).

### Hand preference for food reaching

The median LI (IQR, minimum, maximum) for food reaching was 0.09 (0.31, −0.46, 0.44) for the mothers and −0.06 (0.19, −1, 0.19) for the infants. Five out of seven mother macaques showed a significant hand preference, with two left- and three right-handed individuals. The one-sample sign-tests revealed no significant biases for both mothers ($p = 0.453$, $N = 7$) and infants ($p = 0.453$, $N = 7$). In the case of the infants, six out of seven subjects were involved in the analysis due to the low number of reaching bouts collected for Vanda's infant, that was excluded from the analysis. None of the infants showed a significant hand preference. When comparing the hand preference for food reaching between the two
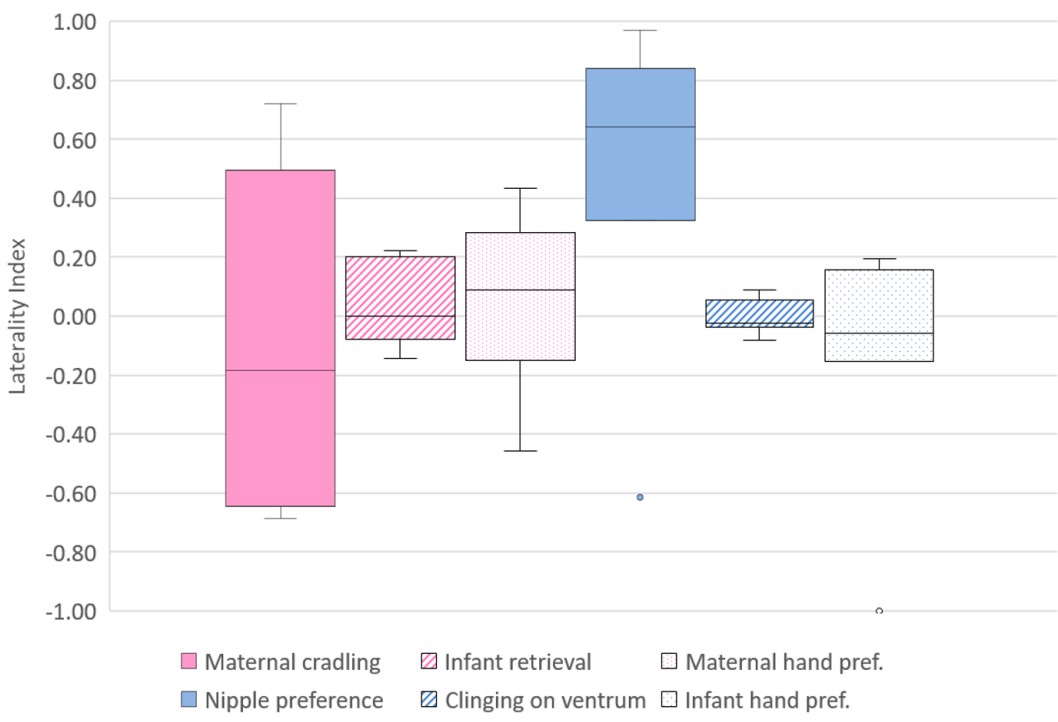

**Figure 1** **Lateral biases in the behaviour of mother and infant Barbary macaques.** Box and whisker plot of the Laterality Index of each behavioural category in mothers (on the left, pale violet red) and infants (on the right, royal blue). The horizontal lines within the box indicate the medians, boundaries of the box indicate the first and third quartile. The whiskers extend up from the top of the box to the largest data element that is less than or equal to 1.5 times the interquartile range (IQR) and down from the bottom of the box to the smallest data element that is larger than 1.5 times the IQR. Values outside this range are considered to be outliers and are drawn as points.

groups, the Mann–Whitney test revealed no significant differences between both the LI ($U = 33.5$; $p = 0.277$) and the ABS-LI ($U = 31.5$; $p = 0.406$).

## Lateral biases in mother and infant behaviour

To investigate differences between the lateral biases in the behaviours of the mothers, the LI and ABS-LI for maternal cradling, infant retrieval and food reaching were compared with each other. Pairwise comparisons by Wilcoxon test were corrected for multiplicity with the Bonferroni-Holm method and revealed no significant difference considering both the LI (Fig. 1) and the ABS-LI (Fig. 2) (see Table 3 for $W$ and $p$ values). For infant lateral biases, we compared both the LI and ABS-LI for clinging on mother's ventrum and food reaching. Wilcoxon test revealed no significant difference when comparing both the LI ($W = 13$, $p = 0.675$) (Fig. 1) and the ABS-LI ($W = 1$, $p = 0.059$) (Fig. 2). Finally, when considering the relationship between nipple and hand preferences of the infants, a significant correlation was found between the LI for nipple preference and food reaching ($rho = 0.786$, $p = 0.048$).

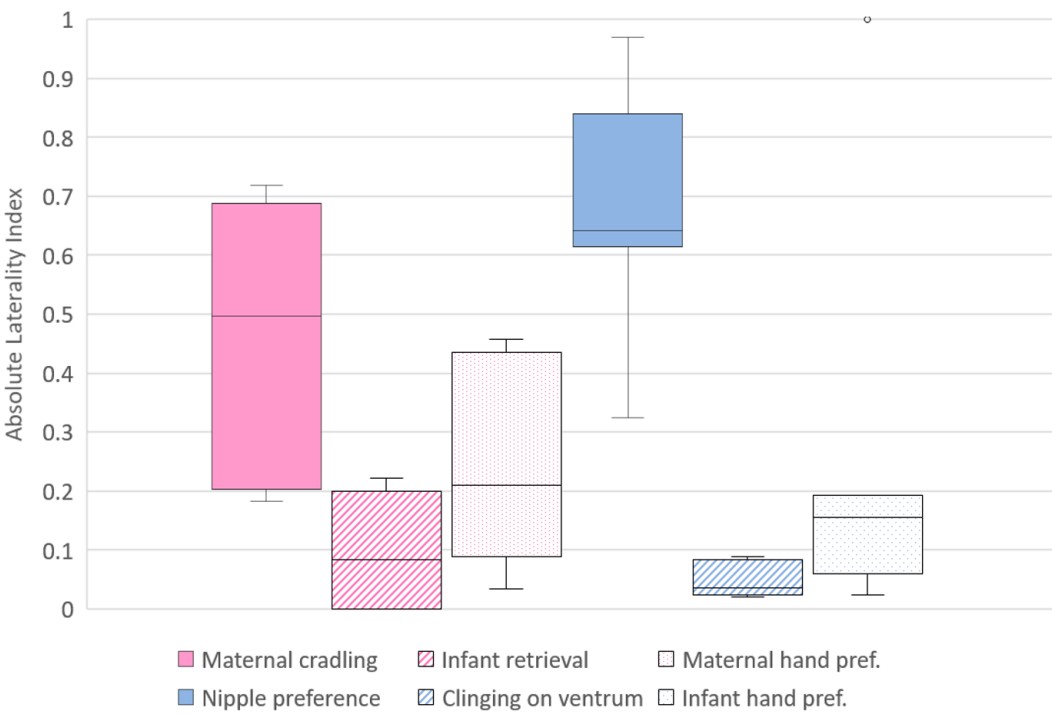

**Figure 2** **Strength of lateral biases in the behaviour of mother and infant Barbary macaques.** Box and whisker plot of the absolute values (ABS) of the Laterality Index of each behavioural category of mothers (on the left, pale violet red) and infants (on the right, royal blue). The horizontal lines within the box indicate the medians, boundaries of the box indicate the first and third quartile. The whiskers extend up from the top of the box to the largest data element that is less than or equal to 1.5 times the interquartile range (IQR) and down from the bottom of the box to the smallest data element that is larger than 1.5 times the IQR. Values outside this range are considered to be outliers and are drawn as points.

## DISCUSSION

Findings from this study highlighted no group-level biases in maternal cradling and infant retrieval by the mothers and the same results emerged for nipple preference and clinging on mother's ventrum in infants Barbary macaques. Regarding maternal cradling, this study agrees with previous literature on macaques, particularly on rhesus macaques (*Tomaszycki et al., 1998*) and Japanese macaques (*Tanaka, 1989*) as well as on other Old-World primates, particularly Sichuan snub-nose monkeys (*Zhao et al., 2008*). Indeed, no lateral bias in cradling infants has been reported in these species. On the other hand, our results disagree with previous studies reporting a left bias in maternal cradling in great apes, chimpanzees and gorillas (*Manning, Heaton & Chamberlain, 1994*; E Toback, 1999, unpublished data) and in humans (*Salk, 1960*; *Damerose & Vauclair, 2002*), suggesting that consistent behavioural lateralization in mother-infant interactions within the Primates order might have first appeared in hominids. However, at the individual level, four out of seven macaques showed negative LI and $z$-scores and three of them had a significant left bias. Thus, lateralization in maternal cradling and the possible presence of a left bias in

**Table 3  Differences between measures of lateral biases within mother Barbary macaques.**

|  | Cradling | Retrieval | Food reaching |
|---|---|---|---|
| **LI** | | | |
| Cradling | # | | |
| Retrieval | $W = 13.5$ <br> $p = 1$ | # | |
| Food reaching | $p' = 1$ <br> $W = 11$ <br> $p = 0.688$ <br> $p' = 2.064$ | $W = 13$ <br> $p = 0.938$ <br> $p' = 1$ | # |
| **ABS-LI** | | | |
| Cradling | # | | |
| Retrieval | $W = 27$ <br> $p = 0.031$ | # | |
| Food reaching | $p' = 0.093$ <br> $W = 23$ <br> $p = 0.150$ <br> $p' = 0.300$ | $W = 5$ <br> $p = 0.156$ <br> $p' = 0.300$ | # |

maternal cradling in this species of macaque as well as in other Old-World monkeys deserve further investigation. Regarding the hand preference for infant retrieval by the mother, no group-level bias was reported. This finding is in agreement with previous studies in rhesus macaques (*Tomaszycki et al., 1998*) and olive baboons (*P. anubis*) (*Damerose & Hopkins, 2002*)

In the case of infant nipple use, the finding of a lack of group-level preference is in agreement with previous research on Japanese macaques (*Hiraiwa, 1981*), pig-tailed macaques (*Erwin, Anderson & Bunger, 1975*) and wild rhesus macaques (*Jaffe et al., 2006*), failing to report a group-level bias in using one nipple or the other one. A similar unbiased distribution of nipple preference was found in other Old-World primates, specifically wild Sichuan snub-nose monkeys (*Zhao et al., 2008*). However, at the individual level six out of seven infant macaques of the current study showed a significant right-nipple preference. Given the small sample size, this high percentage of right biased infants seems to highlight a tendency toward the right nipple. This finding adds consistency to previous research on wild rhesus macaques (*Lindburg, 1971*) in which a bias toward the right nipple was found. However, the reported right nipple preference is not in agreement with the study on rhesus macaques by *Tomaszycki et al. (1998)*, in which a slight group-level left nipple preference was found. Incongruences between studies might be due to differences in sample size, as the current study has a limited sample, and age of the study subjects. Indeed, the age range of the subjects is smaller in the current study than in the previous one by *Tomaszycki et al. (1998)* on rhesus macaques. Indeed, our Barbary macaque infants were less than one month old whereas rhesus macaques were observed from birth until they were six weeks old. The individual-level preference for the mother's right nipple seems partially to resemble the overall tendency to keep the mother on the left side reported in several

non-primate species, particularly bovids, cervids, equids, macropods and marine mammals (*Karenina et al., 2017*; *Giljov, Karenina & Malashichev, 2018*). However, as infant primates tend to suckle frontally on the mother's ventrum and nipples are in the middle of the chest, caution is needed before speculating about this result. Further studies on larger samples of animals are needed, considering the potential impact of factors such as age and species on the development of lateral biases in infant primates. Moreover, there might be other possible explanations for the inconsistencies between studies on lateral biases in mothers and infants. For example, it is possible that not all Old-World monkey species share similar mechanism for nipple preference and different housing conditions between studies might also affect the results (*Jaffe et al., 2006*), as described for other lateralized behaviour (e.g., handedness) (*MacNeilage, 2007*). As reported for maternal cradling, our findings are in contrast with previous studies on chimpanzees and bonobos (*Nishida, 1993*; *Hopkins & De Lathouwers, 2006*), highlighting a left-nipple preference in these species, suggesting differences in lateralization of mother-infant interactions between Old World monkeys and great apes. These findings seem to support the hypothesis relating taxonomic differences in maternal cradling bias and nipple preference to differences in hand preference. In particular, *Hopkins (2004)* suggested that great apes such as chimpanzees and bonobos showing a left-side bias for cradling and nipple preference tend to have a more pronounced right-hand preference, whereas species with right or no bias in mother-infant interactions tend to be left-handed or ambi-preferent.

In addition, we investigated the hand preference for food reaching in both mother and infant macaques. At the group level, no bias in hand preference was found, either for the mothers nor for the infants. This finding agrees with other studies on Barbary macaques, revealing no group-level hand preference on simple food reaching tasks (*Schmitt et al., 2008*; *Regaiolli, Spiezio & Hopkins, 2018*). Moreover, no significant differences were found between mothers and infants in both the LI and the ABS-LI scores, suggesting that mother and infant hand preference is similar in terms of both direction and strength. However, at the individual level, five out of seven mother macaques showed a significant hand preference, whereas no infant was significantly lateralized. This finding seems partially to support the hypothesis that manual lateralization in non-human primates might be affected by the age of the subjects, with older individuals showing a more pronounced hand preference than juveniles (*Warren, 1977*; *Lilak & Phillips, 2008*; *Meguerditchian, Molesti & Vauclair, 2011*). Given the small sample size and the lack of significant differences at the group level, this conclusion is rather speculative and more studies are needed to test the effect of age on the hand preference and to compare manual lateralization between mothers and their infants.

We further investigated differences in the manual lateralization for different behavioural categories involving the use of the hands, considering both direction and strength of the biases. Among the mothers, no significant differences between any measure of lateral bias was found. Similar findings were reported for the infants, showing no significant differences in both the LI and ABS-LI for clinging on mother's ventrum and food reaching.

In addition, to verify the association between the infants' suckling bias and their manual lateralization, a correlation with the LI for nipple preference and food reaching was run,

revealing a slightly positive correlation between the two behavioural categories. Therefore, the position of the nipple chosen to nurse seems to affect the hand preference to reach for food of the infant. A possible explanation could be that the position of the nipple might affect the hand used to hold on the mother body side. For example, if the infants are suckling on the mother right nipple, which is on the left side with respect to the nursing infants, they could be more comfortable to cling on the mother fur on their left, using the ipsilateral hand. Having a nipple preference might therefore lead to the specific and routine use of one hand for support that may persist also outside the nursing and maternal context. This result seems partially to support the Postural Origin Theory, suggesting a left-hand involvement for posture related activities and the right-hand availability for other tasks (*MacNeilage, 2007*). Similar influence of infant early bias on hand preference has been reported in capuchin monkeys (*Cebus apella*). In this species, early bias in head orientation seemed to be related to a body weight displacement of the infant, leading to a manual lateralization for grasping to the mother and to the hand preferences later in development (*Westergaard, Byrne & Suomi, 1998*). Taken together, our findings seem to underline that the maternal environment and anatomy rather than the mother behaviour and side biases would affect the development of handedness in the infants.

## CONCLUSION

The results of this study showed that at the individual level, infant Barbary macaques showed a distinct nipple preference and similar findings have been found for maternal cradling in mother macaques. However, at the group level, no significant biases were found for any of the behavioural categories considered. This lack of group-level side biases in both the mothers and the infants, specifically for maternal cradling and nipple preference, is not in agreement with previous research on great apes. This discrepancy between studies might be due to taxonomic differences in the infant development and interaction with the mother that might affect the handedness. In other words, the influence of maternal behaviour on the infant lateralization reported in great apes and humans might have appeared late in the phylogeny of primates. However, as suggested by our finding in Barbary macaques, the maternal environment and early choice characterizing the life of the infants might affect their hand preference later in development. The differences in lateral biases in maternal and infant behaviour between monkeys and great apes might also explain incongruences between studies on handedness between the two groups. Indeed, some evidence of population-level right handedness has been frequently reported in great apes (e.g., *Meguerditchian et al., 2015*; *Regaiolli, Spiezio & Hopkins, 2016*) but rarely in monkeys, especially during spontaneous unimanual tasks (e.g., *Fitch & Braccini, 2013*; *Regaiolli, Spiezio & Hopkins, 2016*). Overall, our study seems to support the hypothesis that maternal environment and anatomy in early life might affect the development of hand preference in non-human primates (*Hopkins, 1994*; *Hopkins, 1995*; *Westergaard, Byrne & Suomi, 1998*; *Hopkins, 2004*). However, due to the small sample size of the current work and the age differences between different studies, further research on a larger number of mother-infant dyads is needed, in Barbary macaques as well as in other species.

## ACKNOWLEDGEMENTS

We would like to thank Dr. Cesare Avesani Zaborra and Camillo Sandri for allowing this study to take place in Parco Natura Viva. Furthermore, special thanks should be given to Ginevra Rossi and Sebastiano Salvidio for their important contribution to the study design and data collection.

### Funding

The authors received no funding for this work.

### Competing Interests

Barbara Regaiolli is employed by Parco Natura Viva as researcher in the Research and Conservation Department. Caterina Spiezio is employed by Parco Natura Viva as head of the Research and Conservation Department.

### Author Contributions

- Barbara Regaiolli and Caterina Spiezio conceived and designed the experiments, performed the experiments, analyzed the data, prepared figures and/or tables, authored or reviewed drafts of the paper, approved the final draft.
- William Donald Hopkins conceived and designed the experiments, analyzed the data, authored or reviewed drafts of the paper, approved the final draft.

### Animal Ethics

The following information was supplied relating to ethical approvals (i.e., approving body and any reference numbers):

The study was carried out through the live observation of spontaneous behaviours of macaques in their social context. No invasive or stressful techniques were used and the study procedure was in accordance with the EU Directive 2010/63/EU for animal research and the Italian legislative decree 26/2014 for Animal Research.

### Data Availability

The raw data are provided in the Supplemental File and in (Table 1).

### Supplemental Information

Supplemental information for this article can be found online at http://dx.doi.org/10.7717/peerj.4736#supplemental-information.

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
