# Peer review of "Asymmetries in mother-infant behaviour in Barbary macaques (Macaca sylvanus)"

_PeerJ, doi:10.7717/peerj.4736_

## Round 0.1 · original submission · Major Revisions

This is a well presented and interesting manuscript. The reviewers and I feel that the study is certainly publishable but needs some revision. Although the changes are relatively few and small, I have classified the revision as 'major' because one involves statistical issues that could change a conclusion.

Both reviewers have provided helpful comments. Please give careful thought to all of them. Below, I provide my perspective on a few of these comments.

I agree with Reviewer 1 that you should describe in more detail your criteria for considering two events as separate as opposed to part of the same ongoing single event.

Also, I think that Reviewer 1 is correct in indicating that the Spearman correlation matrix requires a correction for multiple tests. It may be advisable to check with a biostatistician as to the best way to achieve such a correction. It is my understanding that Bonferroni correction is too conservative.

As for the use of parametric statistics to describe group-level patterns, I understand that it is possible that the group-level patterns may meet the normality criteria even if the individual-level patterns do not. However, you should provide an indication that the data meet the assumptions for parametric measures or switch to non-parametric measures of central tendancy and variability.

I found your Introduction to the literature quite clear. Give some thought to Reviewer 2's suggestion to summarize the information in a table, but I would accept a text presentation as you have in the present version.

I have provided a pdf with minor language corrections indicated by highlights and inserted comments.

Please check your references carefully (not just the ones I marked in the pdf). Some journal titles have capitals on all words. At least one volume number is missing. Italics and use of abbreviation for journal titles are inconsistent. Hopkins 2007 seems to be indicated as both a book author and editor. Some authors are missing (L410). The full range of pages may be missing from one reference.

In addition, please consider the following suggestions for minor changes:
L33. Is there a good reason why the Abstract does not mention the individual cradling bias?
L112-114. Statement of knowledge gap seems out of place at end of objectives and largely redundant to L101-103.
L121. Age information is redundant to L136 where it seems more appropriate.
L130. It is unclear what 'were not used to interact with humans' means because they were in captivity and presumably had caretakers that provided food, veterinary services, and cleaning and were exposed to the public.
L144, 155. Should you use 'unambiguous' rather than 'unbiased'? I don't see a potential bias here.
L183, 188, 305 (and elsewhere) 'individual level' and similar expressions should have a hyphen only when used as an adjective. (For example, L31, 161, 165 have correct use of hyphen.)
Fig. 1 caption. Do you mean that the error bars represent plus and minus 1 SD?

Reviewer 1 ·

Basic reporting

I believe the manuscript is self-contained and provide relevant results to hypotheses. However, I have a number of specific concerns.

Lines 170-171: One or two citations after “e.g.” would be enough here. Otherwise, it looks like you cite all the studies which use LI scores, while there are a great many of them.
Line 231: “…consistent behavioural lateralization in mother - infant interactions might have first appeared in hominids…” This sounds strange in light of recent studies showing pronounced lateralization of mother - infant interactions in at least 14 (!) non-primate mammal species. Please see the references below.
Karenina K, Giljov A, Ingram J, Rowntree VJ, Malashichev Y. 2017 Lateralization of mother–infant interactions in a diverse range of mammal species. Nature Ecol. Evol. 1, 0030.
Giljov A, Karenina K, Malashichev Y. 2018 Facing each other: mammal mothers and infants prefer the position favouring right hemisphere processing. Biol. Lett.14: 20170707.
Karenina K, Giljov A, De Silva S, Malashichev Y. Social lateralization in wild Asian elephants: visual preferences of young and mothers. Behavioral Ecology and Sociobiology, 72, 21, 2018.
Karenina K, Giljov A, Malashichev Y. Lateralization of mother-infant interactions in wild horses. Behavioural Processes, 148, 49-55, 2018

You cited only one of these studies without any mention of what has been shown. This is not necessary to cite and discuss all the studies on non-primates, but at least Karenina et al., 2017 should be cited and mentioned in the text since it first shows that lateralized mother-infant interactions emerged long before primates. In addition, you may consider a brief discussion of Giljov et al., 2018 since it shows lateralized behaviour of walruses closely resembling cradling in primates.
For reader’s more complete understanding of the topic, it would be useful to mention (at least briefly) the suggested perceptual origin of lateralization in mother-offspring interactions. You investigate the impact of mother-infant lateralized positioning on motor biases, but the influence of sensory modalities should not be neglected. The role of sensory lateralization has been argued in the studies on non-primate mammals (Karenina et al., 2017). Moreover, recent studies of human infant cradling/holding bias (poorly cited, by the way) support the perceptual origin of this kind of lateralization (e.g., Todd, B. K. & Banerjee, R. Lateralization of infant holding by mothers: A longitudinal evaluation of variations over the first 12 weeks. Laterality 21, 12–33 (2016)).
Authors may also consider discussing other kind of lateralized social behaviour of primates and humans - embraces, a kind of 'adult cradling'.
Boeving, E. R., Belnap, S. C., & Nelson, E. L. (2017). Embraces are lateralized in spider monkeys (Ateles fusciceps rufiventris). American Journal of primatology, 79(6).
Packheiser, J., Rook, N., Dursun, Z., Mesenhöller, J., Wenglorz, A., Güntürkün, O., & Ocklenburg, S. (2018). Embracing your emotions: affective state impacts lateralisation of human embraces. Psychological research, 1-11.

Experimental design

Lines 148-150: This may be an important confounding factor since suckling bouts may happen in a sequence (series). That is, behaviourally there was one suckling event (i.e. an infant approached the mother’s tit for suckling), but it was stopped and started again several times for some reasons. In that case, you obtained a series of data which were not independent from each other. The series of data should not be mixed with separate (independent) data points. From your description, it seems to me that you recorded as separate data points both the contacts with the nipple during one suckling interrupted by a few seconds pause and different suckling events between which an infant was involved in other activities or even was away from its mother. From statistical point of view this is not particularly correct. Analysis of separate approaches to mother’s tit for suckling would be much more preferable here, to my opinion. I suggest this cannot be changed already, but try at least to describe the data collection technique in more detail.
Lines 152-154: Were the sequences of reaching events properly recorded and analysed (e.g. several reaching events when sitting at one feeding point and in the same posture were considered as one sequence, then a subject changed location and a new sequence began)? Please describe, how did you separate independent reaching events? I believe that you cannot just record every single case of reaching, but there should be some clear criteria to separate the events. Otherwise, the degree of independence between the data points will vary significantly.

Validity of the findings

Lines 183, 188, 192 and 202: Given the authors used non-parametric tests, none of which is using mean LI's, the reporting of mean LI seems to be redundant. In my opinion, median values would be more appropriate here, because of non-parametric distribution of data. The same applies to Figure 1.
Lines 185, 194 and 204: Given a small sample size (7 subjects in both cases) I recommend authors avoid reporting binomial tests here and simply stick to the existing statements of individual laterality distribution.
Lines 216-221: Threshold levels of significance for correlation coefficients must be adjusted for multiple comparisons by Bonferroni's correction. Will be there a significant correlation between the LI scores for nipple preference and food reaching after this correction? May be need to reconsider also the discussion (lines 286-292).

Additional comments

Authors are recommended to provide more discussion of their findings in the context of recent studies of lateralized mother-infant interactions in both primates and non-primate mammals. The widely assumed association between lateralized infant’s position and perceptual lateralization should be mentioned. To my opinion, the data recording protocol lack standardization in terms of distinguishing of separate behavioural events. I suggest describing the data recording procedure in more detail. If Bonferroni's corrections were made this should be clearly stated, but if it was not made the validity of these results is questionable.

Reviewer 2 ·

Basic reporting

no comment

Experimental design

no comment

Validity of the findings

no comment

Additional comments

(1) In the Introduction, considering the increasing evidences on primate mother-infant behavioral asymmetries, it is better to present one Table containing the basic information on all previous related findings.

(2) In the Methods, on line 162-163, it is better to show how to identify the hand preference when the z score equals 1.96.

(3) In the Discussion, on line 248-249, it is better to show clearly what is the "previous one on rhesus macaques" and what about the age range in that study.

(4) In the Discussion, it is better to make more explanation on the consistency between findings in this study and that in previous studies (e.g. line 225-228, line 237-240).

(5) In the Discussion, it is better to cite the findings on hand preference for other behavioral tasks in the focal species (e.g. Regaiolli et al. Hand preference on unimanual and bimanual tasks in Barbary macaques (Macaca sylvanus). American Journal of Primatology. DOI: 10.1002/ajp.22745).

---

## Round 0.2 · Minor Revisions

The manuscript has been appropriately revised, but a couple of small issues remain.

L184ff. As we discussed in a separate email, the Data Analysis section needs reorganization so that it is clear which analyses were non-parametric and that the use of the z-score, a parametric measure, is justified either by the normal distribution of the data to which it is applied or references indicating that this is the standard approach of the field.

As we discussed in the email, throughout the text and figures, the SD should be replaced by the interquartile range (and possibly maximum and minimum) whenever the median is appropriate, so that the descriptive data elements are consistently non-parametric.

L66. heart, not hearth

L173, 268 Is ‘tit’ correct in this context? Most of your references and the rest of your text use ‘nipple’.

For Fig. 1, the ordinate label should be Laterality Index. It is clearer to spell it out. Specifying median is not appropriate since you will also be showing interquartile range and possibly maximum and minimum. Also, spell out laterality index rather than using the abbreviation in the caption. I think that the different categories of response would be more appropriately called ‘context’ than ‘measures’.

For Fig. 2, please make changes equivalent to those in Figure 1.

For Table 1, the heading should explain the horizontal line in the table.

For Table 2, the heading is inadequate. It should explain more fully what the table is about and identify all abbreviations. It should be clear what the numbers under the R and L refer to (number of events). The test on which the p-value is based should be specified. How does the footnote regarding a binomial test relate to the z-score and p-value?

---

## Round 0.3 · accepted · Accept

The manuscript is now ready for publication, based on the changes and emails with the corresponding author concerning changes to the figure captions.

Minor corrections (which can be performed while in Production):

L173. Extra 'the'; should read 'removed from it'

L370. Capitalize 'We'

Revised figure captions, agreed to by author and academic editor:
Figure 1: Lateral biases in the behaviour of mother and infant Barbary macaques. Box and whisker plot of the Laterality Index of each behavioural category in mothers (on the left, pale violet red) and infants (on the right, royal blue). The horizontal lines within the box indicate the medians, boundaries of the box indicate the first and third quartile. The whiskers extend up from the top of the box to the largest data element that is less than or equal to 1.5 times the interquartile range (IQR) and down from the bottom of the box to the smallest data element that is larger than 1.5 times the IQR. Values outside this range are considered to be outliers and are drawn as points.

Figure 2: Strength of lateral biases in the behaviour of mother and infant Barbary macaques. Box and whisker plot of the absolute values (ABS) of the Laterality Index of each behavioural category of mothers (on the left, pale violet red) and infants (on the right, royal blue). The horizontal lines within the box indicate the medians, boundaries of the box indicate the first and third quartile. The whiskers extend up from the top of the box to the largest data element that is less than or equal to 1.5 times the interquartile range (IQR) and down from the bottom of the box to the smallest data element that is larger than 1.5 times the IQR. Values outside this range are considered to be outliers and are drawn as points.

When revising the captions, the author also revised the figures. The revised figures are acceptable.

#